# CONVOLUTIONAL NEURAL NETWORKS COMBINED WITH RUNGE-KUTTA METHODS

## ABSTRACT

A convolutional neural network for image classification can be constructed mathematically since it can be regarded as a multi-period dynamical system. In this paper, a novel approach is proposed to construct network models from the dynamical systems view. Since a pre-activation residual network can be deemed an approximation of a time-dependent dynamical system using the forward Euler method, higher order Runge-Kutta methods (RK methods) can be utilized to build network models in order to achieve higher accuracy. The model constructed in such a way is referred to as the Runge-Kutta Convolutional Neural Network (RKNet). RK methods also provide an interpretation of Dense Convolutional Networks (DenseNets) and Convolutional Neural Networks with Alternately Updated Clique (CliqueNets) from the dynamical systems view. The proposed methods are evaluated on benchmark datasets: CIFAR-10/100, SVHN and ImageNet. The experimental results are consistent with the theoretical properties of RK methods and support the dynamical systems interpretation. Moreover, the experimental results show that the RKNets are superior to the state-of-the-art network models on CIFAR-10 and on par on CIFAR-100, SVHN and ImageNet.

## 1 INTRODUCTION

Residual Networks (ResNets) which are feed-forward network models with skip connections have achieved great success on several vision benchmarks (He et al., 2016a). Recently, researchers have studied the relation between ResNets and dynamical systems (Liao & Poggio, 2016; E, 2017; Haber et al., 2017; Chang et al., 2018a;b; Lu et al., 2018). Forward Euler method, a first-order RK method, has been employed to explain ResNets with full pre-activation (He et al., 2016b) from the dynamical systems view (Haber et al., 2017; Chang et al., 2018b). Nevertheless, there is no firm evidence to prove that the residual block is just forward Euler method but not any other RK method. We regard the residual mapping as an approximation to the increment in a time-step. The accuracy of the approximation is determined by the structure of the convolutional network. Wide residual network (WRN) (Zagoruyko & Komodakis, 2016) has been proposed to improve the ability of the convolutional subnetwork. However, it is not very efficient only to widen the subnetwork. The new explanation of pre-activation ResNet and its variants which focus on improving residual mapping is one of our contributions.

In addition, some improvements on network architecture based on ordinary differential equations (ODEs) are proposed (Chang et al., 2018a; Lu et al., 2018; Chen et al., 2018). Under the assumption that pre-activation ResNet is forward Euler method, Chang et al. (2018a); Lu et al. (2018) use special linear multi-step methods (LM methods) with low order to construct the network. Chen et al. (2018) utilize a third-party package which offers numerical ODE methods to replace residual block. There is no efficient network architecture for systematic generalization to high order till now. Nevertheless, a higher-order method can achieve a lower truncation error. Since a lower truncation error likely leads to a high accuracy, it is necessary to study an efficient network architecture with a high order.

If the process of image classification is deemed a sequence of time-dependent dynamical systems, there should be a series of ODEs to describe these systems. RK methods are widely-used procedures to solve ODEs in numerical analysis (Butcher, 2008). They are also the building blocks of high-order LM methods. Consequently, these methods can be used to build network models for visual processing.

The neural network community has long been aware of the numerical methods for dynamical systems. Runge-Kutta Neural Network (RKNN) is proposed for identification of unknown dynamical systems in high accuracy (Wang & Lin, 1998), but it has not been used to model the visual system nor been extended to convolutional networks. Moreover, RKNNs adopt a specific RK methods by indicating every coefficient for the RK methods. Thus, it is hard to apply high order RK methods in RKNNs. In addition, the time-step size need to be prespecified. Hence, RKNN cannot be used in tasks where the total time is unknown such as image classification. In contrast, we learn all the coefficients and time-step sizes implicitly by training in order to avoid these difficulties. As a result, one of the major contributions of the paper is a novel and effective neural network architecture inspired by the RK methods.

In order to apply RK methods to the image classification problem, the following assumptions are made throughout the paper. Firstly, the image classification procedure is multi-period and there are transitions between adjacent periods. Secondly, each period is modeled by a time-dependent first-order dynamical system. Based on these assumptions, a novel network model called the RKNet is proposed.

In an RKNet, a period is composed of iterations of time-steps. A particular RK method is adopted throughout the time-steps in a period to approximate the increment in each step. The increment in each step is broken down to the increments in several stages according to the adopted RK method. Each stage is approximated by a convolutional subnetwork due to the versatility of neural networks on approximation.

Another contribution of this paper is a theoretical interpretation of DenseNets and CliqueNets from the dynamical systems view. The dense connections in DenseNet resemble the relationship among increments in the stages in explicit RK methods (ERK methods). Similarly, the clique blocks in CliqueNets resemble the relationship among increments in the stages in implicit RK methods (IRK methods). Under some conditions, DenseNets and CliqueNets can be formulated as approximating dynamical systems using multi-stage RK methods. We also propose a method to convert a DenseNet to an explicit RKNet (ERKNet) and a method convert a CliqueNet to an implicit RKNet (IRKNet). Furthermore, DenseNets and CliqueNets have only one time-step in each period, whereas RKNets are more general and can have multiple time-steps in each period.

We evaluate the performance of RKNets on benchmark datasets including CIFAR-10, CIFAR-100 (Krizhevsky, 2009), SVHN (Netzer et al., 2011) and ILSVRC2012 classification dataset (Russakovsky et al., 2015). Experimental results show that both ERKNets and IRKNets conform to the mathematical properties. Additionally, RKNets achieve higher accuracy than the state-of-the-art network models on CIFAR-10 and comparable accuracy on CIFAR-100, SVHN and ImageNet.

The rest of the paper is organized as follows. The related work is reviewed in Section 2. The architecture of RKNets, the dynamical systems interpretation of DenseNets and CliqueNets, and the conversion from them to RKNets are described in Section 3. The performance of RKNets is evaluated in Section 4. The conclusion and future work is described in Section 5.

## 2 RELATED WORK

ResNets have gained much attention over the past few years since they have obtained impressive performance on many challenging image tasks, such as ImageNet (Russakovsky et al., 2015) and COCO object detection (Lin et al., 2014). ResNets are deep feed-forward networks with the shortcuts as identity mappings. ResNets with pre-activation can be regarded as an unfolded shallow RNN, which implements a discrete dynamical system (Liao & Poggio, 2016). It provides a novel point of view for explaining pre-activation ResNets from dynamical systems view.

Recently, more work has emerged to connect dynamical systems with deep neural networks (E, 2017) or ResNets in particular (Haber et al., 2017; Chang et al., 2018a;b; Li et al., 2018; Long et al., 2018; Lu et al., 2018; Wang et al., 2018; Chen et al., 2018). E (2017) proposes to use continuous dynamical systems as a tool for machine learning. Chang et al. (2018a) propose three reversible architectures based on ResNets and ODE systems. Chang et al. (2018b) propose a novel method for accelerating ResNets training based on the interpretation of ResNets from dynamical systems view (Haber et al., 2017). Li et al. (2018) present a training algorithm which can be used in the context of ResNets. Lu et al. (2018) propose a 2-step architecture based on ResNets. In addition, research

combining dynamical system identification and RK methods with neural networks for scientific computing has emerged recently (Raissi et al., 2017a;b; Raissi, 2018), introducing physics informed neural networks with automatic differentiation. Chen et al. (2018) utilize a third-party package which offers some numerical methods to compute the numerical solution in each time-step.

DenseNets (Huang et al., 2017) are the state-of-the-art network models after ResNets. The dense connection is the main difference from the previous models. There are direct connections from a layer to all subsequent layers in a dense block in order to allow better information and gradient flow. There is no interpretation of DenseNets from dynamical systems view yet. CliqueNets (Yang et al., 2018) are the state-of-the-art network models based on DenseNets. They adopt the alternately updated clique blocks to incorporate both forward and backward connections between any two layers in the same block. However, there is no interpretation of CliqueNets from dynamical systems view yet.

Given that the process of image classification is regarded as a sequence of time-dependent dynamical systems, there should be a set of ODEs that describes these systems. Consequently, mathematical tools could be employed to construct network models. RK methods are commonly used to solve ODEs in numerical analysis (Butcher, 2008). Higher order RK methods can achieve lower truncation error. Moreover, these methods are usually the building blocks of high-order LM methods. Therefore, RK methods are ideal tools to construct network models from dynamical systems view.

RK methods have been adopted to construct neural networks, which are known as RKNN, for identification of unknown dynamical systems described by ODEs (Wang & Lin, 1998). In that paper, neural networks are classified into two categories: (1) a network that directly learns the state trajectory of a dynamical system is called a direct-mapping neural network (**DMNN**); (2) a network that learns the rate of change of system states is called a **RKNN**. Hence, AlexNet (Krizhevsky et al., 2012), VGGNet (Simonyan & Zisserman, 2015), GoogLeNet (Szegedy et al., 2015) and ResNet (He et al., 2016a) all belong to DMNNs. Specifically, the original ResNet (He et al., 2016a) is a DMNN because of the ReLU layer after the addition operation. As a result, the ResNet building block learns the state trajectory directly, not the rate of change of the system states. On the contrary, a ResNet with pre-activation (He et al., 2016b) is an RKNN.

RKNNs are proposed to eliminate several drawbacks of DMNNs, such as the difficulty in obtaining high accuracy for the multi-step prediction of state trajectories. It has been shown theoretically and experimentally that the RKNN has higher prediction accuracy and better generalization capability than the conventional DMNN (Wang & Lin, 1998).

Therefore, it is reasonable to believe that RK methods can be adopted to design effective network architectures for image classification problems. Additionally, the RK methods might improve the performance of image classification since the convolutional subnetworks are able to approximate the rate of change of the dynamical system states more precisely.

## 3 RKNETS

The introduction to RK methods are in Section 3.1. We describe the overall structure of RKNet in Section 3.2. The structure of subnetwork for increment in each time-step is elaborated on in Section 3.3.

### 3.1 RUNGE-KUTTA METHODS

An initial value problem for a time-dependent first-order dynamical system can be described by the following ODE (Butcher, 2008):

$$\frac{d\boldsymbol{y}}{dt} = f\left(t, \, \boldsymbol{y}(t)\right), \qquad \boldsymbol{y}\left(t_0\right) = \boldsymbol{y}_0. \tag{1}$$

where $\boldsymbol{y}$ is a vector representing the system state. The dimension of $\boldsymbol{y}$ should be equal to the dimension of the dynamical system. The ODE represents the rate of change of the system states. The rate of change is a function of time and the current system state. RK methods utilize the rate of change calculated from the ODE to approximate the increment in each time-step, and then obtain the predicted final state at the end of each step. RK methods are numerical methods originated from

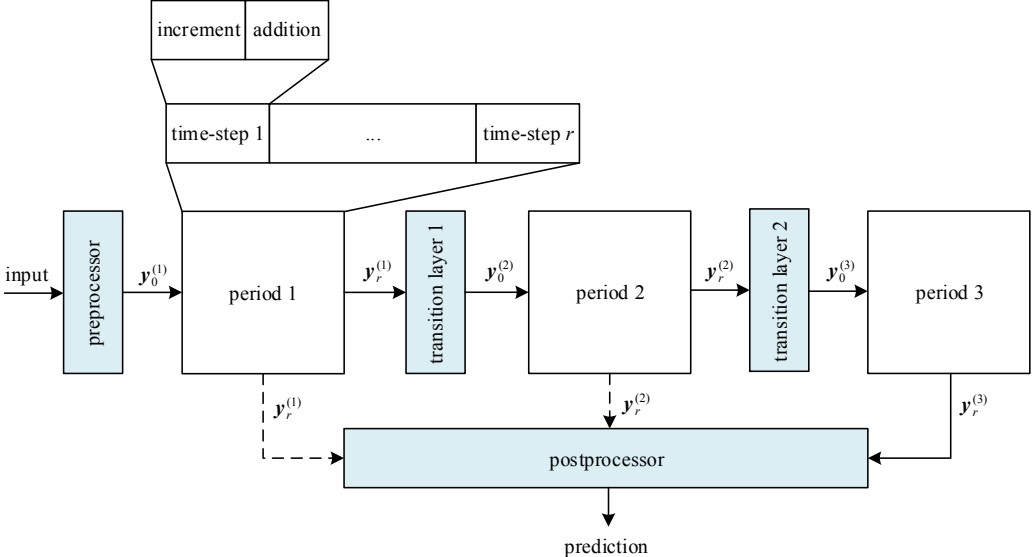

Figure 1: Architecture of a 3-period RKNet. $\boldsymbol{y}^{(d)}$ denotes the system state of period $d$. $\boldsymbol{y}_0^{(d)}$ is the initial state of period $d$. $\boldsymbol{y}_r^{(d)}$ is the final state after $r$ time-steps in period $d$. $r$ is the total number of time-steps in a period. It can vary in different periods. Period 1 and time-step 1 in it are unfolded as an example. System state changes throughout a period. The final state of a step is estimated as the initial state of this step adding an increment. This operation originates from RK methods. To approximate the increment is the key point in RKNet. The dotted lines are for multiscale feature strategy.

Euler method. There are two types of RK methods: explicit and implicit. Both of them are employed in the RKNet. The family of RK methods is given by the following equations (Sli & Mayers, 2003):

$$\boldsymbol{y}_{n+1} = \boldsymbol{y}_n + h \sum_{i=1}^{s} b_i \boldsymbol{z}_i, \qquad t_{n+1} = t_n + h, \qquad (2)$$

where

$$\boldsymbol{z}_i = f\left(t_n + c_i h, \; \boldsymbol{y}_n + h \sum_{j=1}^{s} a_{ij} \boldsymbol{z}_j\right), \qquad 1 \le i \le s. \qquad (3)$$

In equation 2, $\boldsymbol{y}_{n+1}$ is an approximation of the solution to equation 1 at time $t_{n+1}$, i.e. $\boldsymbol{y}(t_{n+1})$; $\boldsymbol{y}_0$ is the input initial value; $h \sum_{i=1}^{s} b_i \boldsymbol{z}_i$ is the increment of system state $\boldsymbol{y}$ from $t_n$ to $t_{n+1}$; $\sum_{i=1}^{s} b_i \boldsymbol{z}_i$ is the estimated slope which is the weighted average of the slopes $\boldsymbol{z}_i$ computed in different stages. The positive integer $s$ is the number of $\boldsymbol{z}_i$, i.e. the number of **stages** of the RK method. The equation 3 is the general formula of $\boldsymbol{z}_i$. $h$ is the time-step size which can be adaptive for different time-steps but must be fixed across stages within a time-step.

In numerical analysis, $s$, $a_{ij}$, $b_i$ and $c_i$ in equation 2 and equation 3 need to be prespecified for a particular RK method. These coefficients are displayed in a Butcher tableau. The ERK methods are those methods with $a_{ij} = 0$ when $1 \le i \le j \le s$. All the RK methods other than ERK methods are IRK methods. The algebraic relationships of the coefficients have to meet the order conditions to reach the highest possible order. Different RK methods have different truncation errors which are denoted by the order: an order $p$ indicates that the local truncation error is $O(h^{p+1})$. If a $s$-stage ERK method has order $p$, then $s \ge p$; if $p \ge 5$, then $s > p$ (Butcher, 2008). Furthermore, a $s$-stage IRK method can has order $p = 2s$ when its coefficients are chosen under some conditions (Butcher, 2008). Therefore, more stages may achieve higher orders, i.e. lower truncation errors. The Euler method is a one-stage first-order RK method with $b_1 = 1$ and $c_1 = 0$. In other words, high-order RK methods can be expected to achieve lower truncation errors than Euler method. Thus, the goal of our proposed RKNets is to improve the classification accuracy by taking advantage of high-order RK methods.

It is necessary to specify $h$ in order to control the error of approximation in common numerical analysis. The varying time-step size can be adaptive to the regions with different rates of change. The truncation error is lower when the $h$ is smaller.

## 3.2 FROM RK METHODS TO RKNETS

There are three components of RKNets: the preprocessor, the multi-periods and the postprocessor. The preprocessor manipulates the raw images and passes the results to the first period. The postprocessor deals with the output from the last period or all the periods while adopting multiscale feature strategy (Yang et al., 2018). Then, it passes the result to the classifier to make a decision. The periods between those two components are divided by the transition layers. These periods can be modeled by time-dependent dynamical systems. Each period of an RKNet is divided into $r$ **time-steps** as shown in Figure 1. RK methods approximate the final state of every time-step using the rate of change of the system state. Some guiding principles when applying RK methods to RKNets are listed as follows.

Firstly, dimensionality reduction is often carried out to simplify the system identification issue, when the dimension of real dynamical system is too high. The dimension of $y$ in each period in RKNet is predefined as the multiplication of the size of feature map and the number of channels at the beginning of a period. The dimensions of $y$ in the same periods of different RKNets can be different due to various degrees of dimensionality reduction. Nevertheless, the dimension of $y$ is consistent within a period.

Secondly, given that there is no explicit ODE for image classification, a convolutional subnetwork is employed to approximate the increment in each time-step. The number of neurons in each hidden layer can be more than the dimension of $y$.

Thirdly, the number of stages $s$ in each period is predefined in RKNet but the other coefficients, $a_{ij}$, $b_i$ and $c_i$ in equation 2 and equation 3 are learned by training. Due to the order conditions (Butcher, 2008), the relationship among the coefficients are more important than the specific value of any individual coefficient. Hence, the coefficients are learned implicitly but not as explicit parameters. The optimal relationship among the coefficients with a highest possible order is obtained after training.

Lastly, the number of time-steps $r$ in each period is predefined in RKNet, but the step size $h$ is learned by training. $n$ in equation 2 and equation 3 is limited to the range $[0, r)$. The learned $h$ is thus considered adaptive. In theory, the adaptive time-step size can achieve higher accuracy.

A variety of RK methods can be adopted in the different periods of RKNets, but the same RK method is used for all time-steps within one period in an RKNet. The network models are named after the specific method in each period, such as RKNet-3×2_4×1_2×5_1×1. The suffix in the name of an RKNet is composed of several $s \times r$ terms; each stands for the method in corresponding period. The number of such terms equals the total number of periods. $s$ or $r$ can vary in different periods. For example, RKNet-3×2_4×1_2×5_1×1 has four periods: period one has 2 time-steps and each step has 3 stages; period two has 1 time-step and it has 4 stages; period three has 5 time-steps and each step has 2 stages; period four has 1 time-step and it has 1 stage. We use this notation throughout this paper. In addition, ERKNets only adopt ERK methods and IRKNets only adopt IRK methods.

Given an RKNet model, $s$ and $r$ can be modified to construct more variants with the same dimensions in the corresponding periods. In other words, $s$ and $r$ control depth of the network while dimensionality reduction controls the width of the network. More stages, more time-steps and larger dimensions usually lead to higher classification accuracy. However, the complexity of an ODE increases with the increase of dimensions. As a result, the convolutional subnetwork which approximates the increment in a time-step need be more complex for larger dimensions. Hence, the accuracy is also associated with the matching degree of the dimension and the convolutional subnetwork. The unmatched high-dimensional network model may have lower accuracy. Additionally, the training method might affect the classification accuracy too.

## 3.3 ERKNETS AND IRKNETS

In this section, we introduce the architecture of RKNets. As shown in equation 2, the sum of $hb_i z_i$ represents the increment in a time-step. It is crucial to approximate this increment in RKNet. For

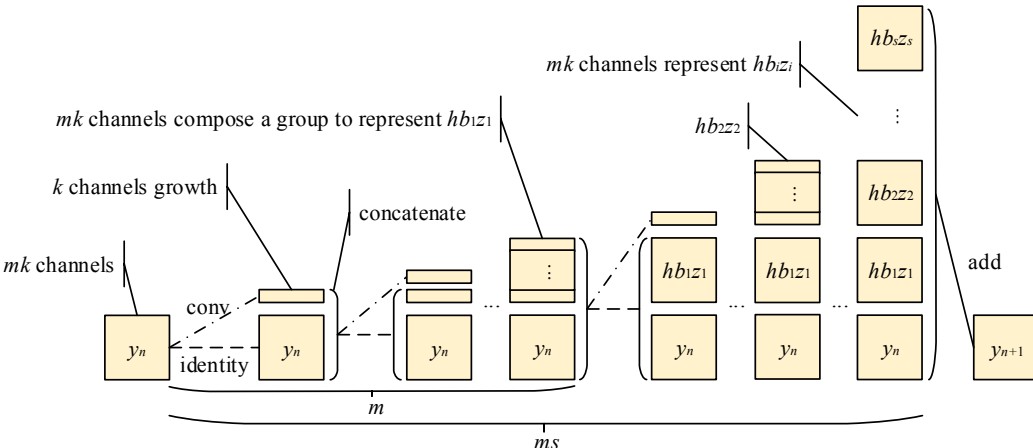

Figure 2: Architecture of one time-step in ERKNet using an $s$-stage ERK method. $\boldsymbol{y}_n$ is the approximation of $\boldsymbol{y}(t_n)$. A dense block grows every $m$ times at a growth rate of $k$ to form a convolutional subnetwork for generating each $hb_i\boldsymbol{z}_i$. Here, $h$ is time-step size, $b_i$ is coefficient of ERK method, and $\boldsymbol{z}_i$ is the slope of each stage in ERK method. The total number of growth is $ms$ in a dense block in order to generate $hb_i\boldsymbol{z}_i$ for $i = 1, \ldots, s$. An explicit summation layer is added after a dense block to complete a time-step.

the purpose of constructing an RKNet, it is necessary to hide the time-step size and the coefficients in RK methods. $hb_i\boldsymbol{z}_i$ can be described as follows according to equation 3:

$$
\begin{aligned}
hb_i\boldsymbol{z}_i &= hb_i f \left( t_n + c_i h, \; \boldsymbol{y}_n + h \sum_{j=1}^{s} a_{ij}\boldsymbol{z}_j \right) \\
&= g_i \left( \boldsymbol{y}_n + h \sum_{j=1}^{s} a_{ij}\boldsymbol{z}_j \right) \\
&= F_i \left( \boldsymbol{y}_n, \; ha_{i1}\boldsymbol{z}_1, \; \ldots, \; ha_{is}\boldsymbol{z}_s \right).
\end{aligned}
\tag{4}
$$

The above transformation first changes the explicit dependence on the time in equation 3 to an implicit one. Since the time parameter $t_n + c_i h$ is different for the different stages, it can be absorbed into $g_i(\cdot)$, which implicitly depends on time for stage $i$. Afterward, the summation in the input parameter of $g_i(\cdot)$ is split into separate terms. $F_i(\cdot)$ denotes the function of these terms for each stage. We verify that $F_i(\cdot)$ can equal to $g_i(\cdot)$ after training by experiment though $F_i(\cdot)$ is more expressive than $g_i(\cdot)$ in expression. Additionally, $F_i(\cdot)$ is more memory efficient than $g_i(\cdot)$ because of saving the storage for the summation inputted to $g_i(\cdot)$.

### 3.3.1 CONNECT ERKNETS WITH DENSENETS

In order to construct ERKNets, $hb_i\boldsymbol{z}_i$ can be described by the equation below, according to equation 4.

$$
\begin{aligned}
hb_i\boldsymbol{z}_i &= e_i \left( \boldsymbol{y}_n, \; ha_{i1}\boldsymbol{z}_1, \; \ldots, \; ha_{i(i-1)}\boldsymbol{z}_{i-1} \right) \\
&= E_i \left( \boldsymbol{y}_n, \; hb_1\boldsymbol{z}_1, \; \ldots, \; hb_{i-1}\boldsymbol{z}_{i-1} \right).
\end{aligned}
\tag{5}
$$

The above transformation first eliminates $ha_{ij}\boldsymbol{z}_j$ $(i \leq j)$ from $F_i(\cdot)$ in equation 4 since $a_{ij} = 0$ when $1 \leq i \leq j \leq s$ for ERK methods (See 3.1). As a result, $hb_i\boldsymbol{z}_i$ is denoted by a function of $y_n$ and $ha_{ij}\boldsymbol{z}_j$ for $j = 1, \ldots, i-1$. It is written as $e_i(\cdot)$. After that, adjusting the coefficients of each parameter from $a_{ij}$ to $b_j$ yields another function $E_i(\cdot)$. It is a function of $y_n$ and $hb_j\boldsymbol{z}_j$ for $j = 1, \ldots, i-1$.

If a convolutional subnetwork is adopted to model $E_i(\cdot)$ in equation 5, the most similar network structure is the dense connections in DenseNets. To be specific, a growth in a dense block concatenates all the preceding layers as the input of convolutional subnetwork just like that $hb_i\boldsymbol{z}_i$ uses $y_n$

and all the increments in preceding stages as the input of $E_i(\cdot)$. For the purpose of adopting dense block in ERKNets, the dense blocks must conform to the following rules.

**Rule 1** The number of channels of $\boldsymbol{y}_n$ is in the form of $mk$, where $m$ and $k$ are positive integers and $k$ is known as the growth rate in DenseNet literature. The dimension of $\boldsymbol{y}_n$ is the multiplication of the size of feature map and $mk$.

**Rule 2** Every $m$ successive growth constructs a convolutional subnetwork for $E_i(\cdot)$. Each subnetwork outputs $mk$ channels which are regarded as a group according to the number of channels of $\boldsymbol{y}_n$. Each convolutional subnetwork concatenates $\boldsymbol{y}_n$ and all the preceding groups as its input. The $i$th group generated by the $i$th subnetwork corresponds to $hb_i\boldsymbol{z}_i$.

**Rule 3** The total number of growth is $ms$, where $s$ is number of stages of RK methods. Consequently, $s$ groups representing $hb_i\boldsymbol{z}_i$ for $i = 1, \ldots, s$ are generated by $s$ convolutional subnetworks modeling $E_i(\cdot)$ for $i = 1, \ldots, s$ successively in a dense block.

Appending to a restricted dense block conforming to the above rules, $\boldsymbol{y}_n$ and the groups $hb_i\boldsymbol{z}_i$ for $i = 1, \ldots, s$ are added to obtain $\boldsymbol{y}_{n+1}$ according to equation 2. Figure 2 illustrates one time-step of ERKNet.

In DenseNets, every dense block together with part of the subsequent computation can be regarded as a period using a $s$-stage ERK method with $r = 1$ time-step. The transition layers and the postprocessor contain the summation operation in equation 2. This gives an explanation of DenseNets from the dynamical systems view.

### 3.3.2 CONNECT IRKNETS WITH CLIQUENETS

$hb_i\boldsymbol{z}_i$ for IRK methods can be described by the equation below, according to equation 4.

$$
\begin{aligned}
hb_i\boldsymbol{z}_i &= H_i\left(\boldsymbol{y}_n,\ hb_1\boldsymbol{z}_1,\ \ldots,\ hb_s\boldsymbol{z}_s\right) \\
&= G_i\left(hb_1\boldsymbol{z}_1,\ \ldots,\ hb_{i-1}\boldsymbol{z}_{i-1},\ hb_{i+1}\boldsymbol{z}_{i+1},\ \ldots,\ hb_s\boldsymbol{z}_s\right) \\
&= I_i\left(hb_1\boldsymbol{z}_1,\ \ldots,\ hb_{i-1}\boldsymbol{z}_{i-1},\ \boldsymbol{v}_{i+1},\ \ldots,\ \boldsymbol{v}_s\right)
\end{aligned}
\tag{6}
$$

where

$$
\begin{aligned}
\boldsymbol{v}_j &= V_j(hb_j\boldsymbol{z}_j) \\
&= J_j\left(\boldsymbol{y}_n,\ \boldsymbol{v}_1,\ \ldots,\ \boldsymbol{v}_{j-1}\right).
\end{aligned}
\tag{7}
$$

The above transformation first adjusts the coefficients of each parameter of $F_i(\cdot)$ in equation 4 from $a_{ij}$ to $b_j$. It yields another function $H_i(\cdot)$. As a result, every $hb_i\boldsymbol{z}_i$ is a function of $\boldsymbol{y}_n$. Thus, $hb_i\boldsymbol{z}_i$ can be denoted by a function of $hb_j\boldsymbol{z}_j$ for $j = 1, \ldots, s, j \neq i$. This function is written as $G_i(\cdot)$.

Inspired by Newton method which is used to implement IRK methods (Butcher, 2008), $hb_i\boldsymbol{z}_i$ is initialized using all available information firstly and then updated alternately. Given $\boldsymbol{v}_j$ is the initial value of $hb_j\boldsymbol{z}_j$, the relationship between them is denoted by the function $V_j(\cdot)$. Therefore, $hb_i\boldsymbol{z}_i$ can be denoted by a function of $hb_j\boldsymbol{z}_j$ for $j = 1, \ldots, i-1$ and $\boldsymbol{v}_j$ for $j = i+1, \ldots, s$. This function is written as $I_i(\cdot)$. It is the update function of $hb_i\boldsymbol{z}_i$. Since every $hb_j\boldsymbol{z}_j$ is a function of $\boldsymbol{y}_n$, every $\boldsymbol{v}_j$ is also a function of $\boldsymbol{y}_n$. Thus, $\boldsymbol{v}_j$ can be denoted by a function of $\boldsymbol{y}_n$ and $\boldsymbol{v}_q$ for $q = 1, \ldots, j-1$. This function is written as $J_j(\cdot)$. It is the initialization function of $hb_j\boldsymbol{z}_j$.

The update process is a sequence of iterations till convergence in Newton method. In other words, $\boldsymbol{v}_j$ is updated for many times to approach $hb_j\boldsymbol{z}_j$. During updating, $G_i(\cdot)$ with the biased input is used as the update function since $I_i(\cdot)$ is unknown. If using convolutional subnetwork to model each $I_i(\cdot)$, these functions can be learned under the help of training. As a result, each $\boldsymbol{v}_j$ needs to be updated only once. Therefore, the computational cost is reduced remarkably.

If a convolutional subnetwork is adopted to model $J_j(\cdot)$ in equation 7 and $I_i(\cdot)$ in equation 6, the most similar network structure is the clique block in CliqueNets. To be specific, a clique block is composed of Stage-I and Stage-II in CliqueNet literature. Stage-I which initializes all layers in a clique block is regarded as a sequence of $J_j(\cdot)$. Then, Stage-II for updating all layers alternately corresponds to all $I_i(\cdot)$. In CliqueNet literature, all layers in a clique block except the top layer to be updated are concatenated as the bottom layer, i.e. the input of a convolutional subnetwork for updating. It is just like $I_i(\cdot)$ uses $hb_j\boldsymbol{z}_j$ for $j = 1, \ldots, i-1$ and $\boldsymbol{v}_j$ for $j = i+1, \ldots, s$ as input. In order to adopt clique block in IRKNets, the clique blocks must conform to the following rules.

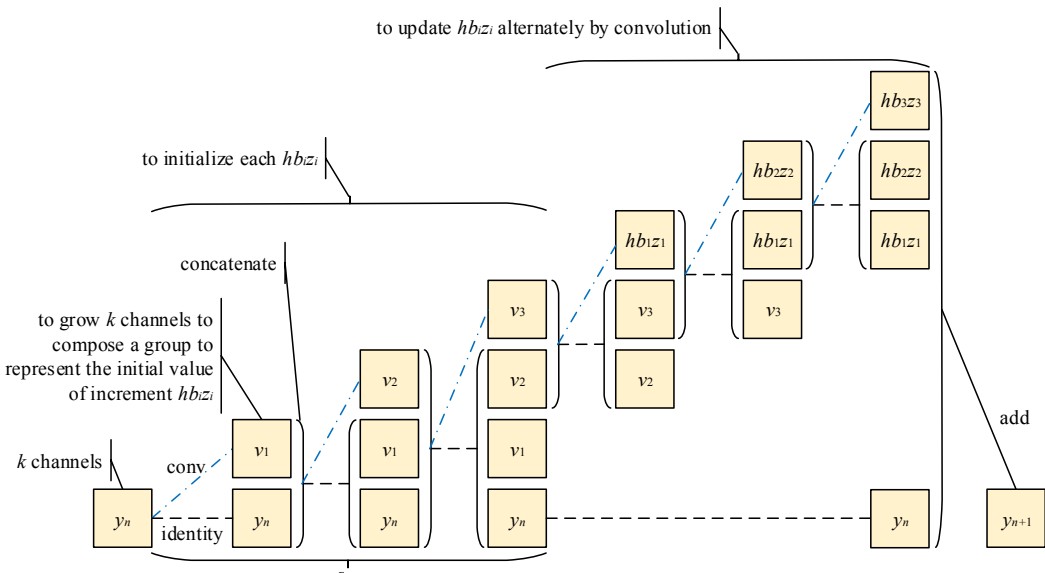

Figure 3: Architecture of one time-step in IRKNet using a 3-stage IRK method. $\boldsymbol{y}_n$ is the approximation of $\boldsymbol{y}(t_n)$. A dense block, which is Stage-I of a clique block, grows $k$ channels every time to generate the initial value of each $hb_i\boldsymbol{z}_i$, written as $\boldsymbol{v}_i$. Here, $h$ is time-step size, $b_i$ is coefficient of IRK method, and $\boldsymbol{z}_i$ is the slope of each stage in IRK method. In Stage-II of a clique block, the convolutional subnetwork concatenating the current values of $hb_j\boldsymbol{z}_j$ for $j = 1, \ldots, 3$, $j \neq i$ to update every $hb_i\boldsymbol{z}_i$ alternately. An explicit summation layer is added after a clique block to complete a time-step.

**Rule 1** The number of channels of $\boldsymbol{y}_n$ is $k$, which is the growth rate in Stage-I since Stage-I is a dense block. The dimension of $\boldsymbol{y}_n$ is the multiplication of the size of feature map and $k$.

**Rule 2** Every growth in Stage-I constructs a convolutional subnetwork. Each subnetwork outputs $k$ channels which are regarded as a group according to the number of channels of $\boldsymbol{y}_n$. Each convolutional subnetwork concatenates $\boldsymbol{y}_n$ and all the preceding groups as its input. The $i$th group generated by the $i$th subnetwork is $\boldsymbol{v}_i$.

**Rule 3** The total number of growth in Stage-I is $s$, which is number of stages of RK methods. Consequently, $s$ groups representing $\boldsymbol{v}_i$ for $i = 1, \ldots, s$ are generated by $s$ convolutional subnetworks successively in Stage-I. $s$ should be larger than 1 for updating alternately in Stage-II.

Appending to a restricted clique block conforming to the above rules, $\boldsymbol{y}_n$ and the groups $hb_i\boldsymbol{z}_i$ for $i = 1, \ldots, s$ are added to obtain $\boldsymbol{y}_{n+1}$ according to equation 2. Figure 3 illustrates one time-step of IRKNet using a 3-stage IRK method as an example.

In CliqueNets, every clique block together with part of the subsequent computation can be regarded as a period using a $s$-stage IRK method with $r = 1$ time-step. The transition layers and the post-processor contain the summation operation in equation 2. This gives an explanation of CliqueNets from the dynamical systems view.

## 4 EXPERIMENTS

To verify the theoretical properties of RK methods and evaluate the performance of RKNets on image classification, experiments are conducted using the proposed network architectures. The experimental setup is described in Appendix A. Some extra techniques, including attentional transition, bottleneck and multiscale feature strategy, can be adopted in RKNets following CliqueNets. They are introduced in Appendix B.

Table 1: Test errors of ERKNets and IRKNets, evaluated on CIFAR-10 without data augmentation. The growth rate $k$ is 36 in every period of the RKNets. The times of successive growth in each stage, $m$, is 1. The multiscale feature strategy is used. All the models are run with batchsize 64.

| ERKNet | FLOPs (G) | Params (M) | Error (%) | IRKNet | FLOPs (G) | Params (M) | Error (%) |
|---|---|---|---|---|---|---|---|
| -6×1_6×1_6×1 | 0.66 | 0.74 | 7.08 | -3×1_3×1_3×1 | 0.38 | 0.32 | 7.18 |
| -7×1_6×1_6×1 | 0.83 | 0.83 | 7.02 | -4×1_3×1_3×1 | 0.62 | 0.40 | 6.89 |
| -7×1_7×1_6×1 | 0.87 | 0.91 | 6.67 | -4×1_4×1_3×1 | 0.68 | 0.49 | 6.63 |
| -7×1_7×1_7×1 | 0.88 | 0.99 | 6.61 | -4×1_4×1_4×1 | 0.69 | 0.57 | 6.50 |

Table 2: Test errors evaluated on CIFAR and SVHN. $k$ is growth rate. The multiscale feature strategy is used in RKNets. A and B represent attentional transition and bottleneck respectively. The bottleneck layers which output $k$ channels to the following layers are used in IRKNets. C10 and C100 stand for CIFAR-10 and CIFAR-100 respectively. "+" indicates standard data augmentation. When data augmentation is not used, dropout layers are added. The values with * are provided by Huang et al. (2017). The values with † are provided by Kuen et al. (2017). The values with ⋆ are computed by ourselves. FLOPs and Params are calculated on CIFAR-10 or SVHN. RKNets are run with batchsize 32 on CIFAR but run with batchsize 64 on SVHN. Results that outperform all competing methods are **bold** and the overall best result is **blue**.

| Model | FLOPs (G) | Params (M) | C10 (%) | C10+ (%) | C100 (%) | C100+ (%) | SVHN (%) |
|---|---|---|---|---|---|---|---|
| pre-act ResNet (He et al., 2016b) | - | 10.2 | 10.56* | 4.62 | 33.47* | 22.71 | - |
| WRN (Zagoruyko & Komodakis, 2016) | 3.10† | 11.0 | - | 4.27 | - | 20.43 | 1.54 |
| | 10.49† | 36.5 | - | 4.00 | - | 19.25 | - |
| DenseNet (Huang et al., 2017) | 14.53⋆ | 27.2 | 5.83 | 3.74 | 23.42 | 19.25 | 1.59 |
| | 10.83⋆ | 15.3 | 5.19 | 3.62 | **19.64** | 17.60 | 1.74 |
| | 18.59⋆ | 25.6 | - | 3.46 | - | 17.18 | - |
| Hamiltonian (Chang et al., 2018a) | - | 1.68 | - | 5.98 | - | 26.11 | - |
| LM-architecture (Lu et al., 2018) | - | 1.7 | - | 5.27 | - | 22.9 | - |
| | - | 68.8 | - | - | - | **16.79** | - |
| CliqueNet (Yang et al., 2018) | 9.45 | 10.14 | 5.06 | - | 23.14 | - | **1.51** |
| | 10.56⋆ | 10.48⋆ | 5.06 | - | 21.83 | - | 1.64 |
| IRKNet-5×1_5×1_5×1-AB ($k$=80) | 2.17 | 1.40 | 5.27 | 4.23 | 24.35 | 21.77 | 1.74 |
| IRKNet-5×1_5×1_5×1-A ($k$=80) | 5.44 | 4.37 | - | - | - | - | 1.63 |
| IRKNet-5×1_5×1_5×1-AB ($k$=150) | 7.62 | 4.87 | **4.60** | 3.60 | 21.39 | 19.42 | 1.64 |
| IRKNet-6×1_6×1_6×1-A ($k$=80) | 7.92 | 6.28 | - | - | - | - | 1.52 |
| IRKNet-5×1_5×1_5×1-AB ($k$=180) | 10.98 | 6.99 | **4.56** | - | 20.88 | 18.61 | - |
| IRKNet-5×1_5×1_5×1-AB ($k$=200) | 13.55 | 8.63 | - | 3.54 | 20.67 | 18.11 | - |
| IRKNet-5×1_5×1_5×1-AB ($k$=240) | 19.51 | 12.41 | - | **3.40** | 20.58 | - | - |

According to the theoretical results, an RK method with more stages usually has a higher order and a lower truncation error. Therefore, as the number of stages increases, a more precise approximation of the system states in every period leads to more accurate classification. Table 1 shows the number of FLOPs and parameters and classification error on CIFAR-10 for RKNets with varying number of stages in each period. The empirical results are consistent with the theoretical properties.

Table 3: Classification errors on ImageNet validation set with a single-crop (224×224). The growth rate $k$ is 32 and $mk$ is the initial number of channels in each period in RKNets. $m_n$ stands for $m$ in the $n$th period. For each RKNet in this table, $m_0$ is 2, $m_1$ is 4 and $m_2$ is 8. B represents bottleneck. The bottleneck layers which output $4k$ channels to the following layers are used in ERKNets.

| Model | $m_3$ | FLOPs (G) | Params (M) | Top1 (%) | Top5 (%) |
|---|---|---|---|---|---|
| ERKNet-3×1_3×1_3×1_1×1-B | 16 | 5.20 | 6.95 | 25.47 | 7.81 |
| ERKNet-3×1_3×1_4×1_2×1-B | 20 | 6.35 | 14.49 | 24.12 | 7.17 |
| ERKNet-3×1_3×1_6×1_2×1-B | 28 | 8.50 | 25.51 | 23.14 | 6.66 |

Figure 4: Comparison of the DenseNets, CliqueNets and RKNets. The top-1 error rates (single-crop testing) on the ImageNet validation dataset are shown as a function of learned parameters (left) and FLOPs during test-time (right). RKNets compared here are the models shown in Table 3.

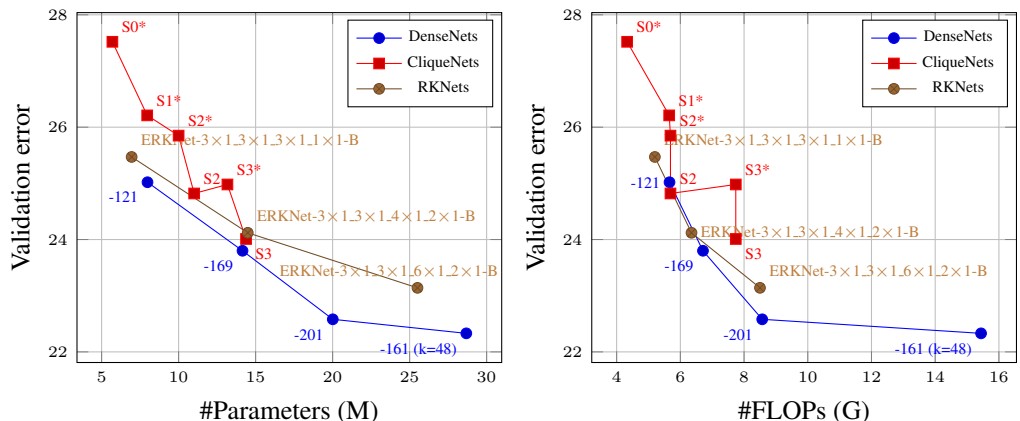

IRKNets are evaluated on CIFAR-10, CIFAR-100 and SVHN while ERKNets are evaluated on ImageNet to compare with the state-of-the-art network models. The test errors of IRKNets on CIFAR-10, CIFAR-100 and SVHN are shown in Table 2. The top-1 and top-5 errors on ImageNet validation set with a single-crop (224×224) are shown in Table 3. Figure 4 shows the single-crop top-1 validation errors of DenseNets, CliqueNets and RKNets as a function of the number of parameters (left) and FLOPs (right). According to the experimental results, RKNets are more efficient than the state-of-the-art models on CIFAR-10 and on par on CIFAR-100, SVHN and ImageNet.

## 5 CONCLUSION

We propose to employ a type of numerical ODE methods, the RK methods, to construct convolutional neural networks for image classification tasks. The proposed network architecture can systematically generalize to high order. At the same time, we give a theoretical interpretation of the DenseNet and CliqueNet via the dynamical systems view. The model constructed using the RK methods is referred to as the RKNet, which can be converted from a DenseNet or CliqueNet by enforcing theoretical constraints.

The experimental results validate the theoretical properties of RK methods and support the dynamical systems interpretation. Moreover, the experimental results demonstrate that RKNets surpass the state-of-the-art models on CIFAR-10 and are on par on CIFAR-100, SVHN and ImageNet.

With the help of the dynamical systems view and various numerical ODE methods including RK methods, more general neural networks can be constructed. Many aspects of RKNets and the dynamical systems view still require further investigation. We hope this work inspires future research directions.

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

## A  EXPERIMENTAL SETUP

The RKNets are evaluated on CIFAR-10, CIFAR-100, SVHN and ImageNet. The CIFAR-10 dataset contains 60,000 color images of size $32 \times 32$ in 10 classes, with 5,000 training images and 1,000 test images per class. The CIFAR-100 is similar to the CIFAR-10 except that it has 100 classes and 500 training images and 100 test images per class. The Street View House Numbers (SVHN) dataset (Netzer et al., 2011) contains $32 \times 32$ colored digit images. There are 73,257 images in the training set, 26,032 images in the test set, and 531,131 images for additional training. ImageNet, which denotes the ILSVRC2012 classification dataset in this paper, consists of 1.28 million training images and 50,000 validation images. It has 1,000 classes and $732 \sim 1,300$ training images and 50 validation images per class.

The weights of convolution layer are initialized as in (He et al., 2015). A weight decay of 0.0001 and Nesterov momentum of 0.9 are used. The learning rate is set to 0.1 initially.

On both CIFAR and SVHN, the learning rate is divided by 10 at 50% and 75% of the training procedure. Moreover, the weights of fully connected layer are using Xavier initialization (Glorot & Bengio, 2010). For the cases without data augmentation, we add a dropout layer (Srivastava et al., 2014) with dropout rate 0.2 after each convolution layer following (Huang et al., 2017; Yang et al., 2018).

On CIFAR, the models are trained using stochastic gradient descent with a mini-batch size of 64 or 32 as required. A standard data augmentation scheme is adopted in some cases following (He et al., 2016a). The models are trained for 300 epochs.

On SVHN, the models are trained using stochastic gradient descent with a mini-batch size of 64. Following (Yang et al., 2018), we use all training samples without augmentation and divide images by 255 for normalization. The models are trained for 40 epochs.

On ImageNet, the models are trained with a mini-batch size of 256 for 90 epochs. Scale and aspect ratio augmentation in (Szegedy et al., 2015), the standard color augmentation in (Krizhevsky et al., 2012) as well as the photometric distortions in (Howard, 2014) are adopted. The learning rate is divided by 10 every 30 epochs.

## B  EXTRA TECHNIQUES

The attentional transition is a channelwise attention mechanism in transition layers, following the method proposed in (Yang et al., 2018). In attentional transition, the filters are globally averaged after the convolution in transition firstly. Then, two fully connected (FC) operations are conducted. The first FC layer has half of the filters and is activated by a ReLU function. The second FC layer has the same number of filters and is activated by a sigmoid function. At last, the output of the second FC layer acts on the output of the convolution by filter-wise multiplication. The bottleneck layer is a $1 \times 1$ convolution layer which is placed before each $3 \times 3$ convolution layer in periods. The multiscale feature strategy is a mechanism in the postprocessor to collect outputs from all the periods but not only from the last period.

