# OpenReview forum: "Convolutional Neural Networks combined with Runge-Kutta Methods"
_ICLR.cc/2019/Conference_

### Official Review · AnonReviewer3 · 2018-11-03
**Review TLDR. Good subject, paper can be presented a bit more clearer**

**Rating:** 6
**Confidence:** 3

**Review:**


Authors describe a method  in which based on the dynamical system description of how a neural networks work, one can construct an analogues process based on Runge Kutta method.

The paper is sound mathematically, touches upon ideas that have been circling around for a while (https://arxiv.org/pdf/1804.04272.pdf for example) and presents a nice test case for integrating tools from dynamical systems into deep learning.

The presentation though is a bit convoluted and unclear and I would strongly encourage the writers to break it down and make it more readable

The method is based on looking at the midsection rules of RK as a feed forward process and mapping that as information propagation in the network. RK methods can be pretty sensitive to Chaos and other non linear effects if one does not take a *very small* time step discretization step. Did you perform some sort of stability analysis on networks as a function of non linear activations inside of it ?
Also RK methods a pretty expensive compared to most standard Suzuki-Trotter, can you remark a bit on the computational aspects of you experiments ?
I think that a big chunk of 3.1 can be either taken out to the appendix or just give a reference to a standard text book and elaborate on training time/resources etc ….

As a generic question I wonder how does this approach handle things which are weak perception or non perception methods, did you try this on any NLP problems or generic tabular data ?

---

> ### Author Response · Authors · 2018-11-26
> **Clarifications and updated paper**
>
> Thank you for your constructive feedback.
>
> -->"The presentation though is a bit convoluted and unclear and I would strongly encourage the writers to break it down and make it more readable"
>
> We have updated the paper following your advice.
>
> -->"RK methods can be pretty sensitive to Chaos and other non linear effects if one does not take a *very small* time step discretization step. Did you perform some sort of stability analysis on networks as a function of non linear activations inside of it ?"
>
> The time-step size should be small enough to ensure the stability but the time-step size in image classification is unknown. However, to stay in the stability region of a method means that the computed solution remains bounded after many steps of computation. In other words, the stability condition results in a precise approximation. Consequently, if the prediction accuracy of the network model is high enough, then the adopted RK methods and the number of time-steps meet the stability condition.
>
> -->"Also RK methods a pretty expensive compared to most standard Suzuki-Trotter, can you remark a bit on the computational aspects of you experiments ?"
>
> There is a high computational cost in computing derivative at each stage in a time-step. However, the total computational cost is the multiplication of the cost in each time-step and the number of time-steps. According to the experimental results in our paper, RKNets adopting only one time-step can obtain the competitive FLOPs efficiency.
>
> -->"I think that a big chunk of 3.1 can be either taken out to the appendix or just give a reference to a standard text book and elaborate on training time/resources etc ."
>
> We have added more discussions to paper, especially on the reason why the computational cost of IRKNet is reduced remarkably compared to the common implementation of IRK methods.
>
> -->"As a generic question I wonder how does this approach handle things which are weak perception or non perception methods, did you try this on any NLP problems or generic tabular data ?"
>
> We have not tried RKNet on other problems yet. Nevertheless, RKNet should be able to handle other dynamical system identification problems. It is worth investigating how to define a dynamical system for a different problem.

---

### Official Review · AnonReviewer2 · 2018-11-03
**Connecting CNNs with ODE solvers**

**Rating:** 5
**Confidence:** 4

**Review:**

The paper takes the perspective that a ResNet can be seen as a discrete time dynamical system approximation, and then uses that insight to introduce a Runge Kutta style continuous dynamical system.  Results are achieved that outperform DenseNet and CliqueNet

Quality
- The connection between deep nets and dynamical systems is an important one, as a small but growing literature demonstrates.
- The quality of this paper is difficult to extract, given rather long and unclear definitions of various key choices, at key points opting instead for a speculative connection to neurosciences (see clarity below).
- The empirical results could make this paper compelling, but the comparisons are rather shallow, in that they compare on only a few datasets and don't compare extensively to many of the state of the art methods.   For example, in Table 2 DenseNet outperforms the IRKNets, but this fact is not highlighted.  Overall this and other key omissions create the concern that the results are not particularly strong.
- One key reference that is missing is the careful work Chen et al https://arxiv.org/abs/1806.07366.  This work provides an excellent basis for how to present these ideas rigorously.

Clarity
- The connections to the brain -- such as "visual cortex", "ventral stream", etc -- are not rigorous and are unrelated to the paper at hand; really all connections to brain should be removed.
- Figures 2 and 3 are the central architectural choices, but the writing around them does not clarify why all the choices have been made, and what the implications are.

Originality
- This is one of a small but growing number of papers connecting deep nets to ODE solvers.  Chen et al should be cited (Chen et al https://arxiv.org/abs/1806.07366).
- There appears to be sufficient originality.

Significance
- hard to determine with unclear exposition and rather incomplete empirical results.

---

> ### Author Response · Authors · 2018-11-26
> **Clarifications and updated paper**
>
> We highly appreciate your constructive comments and the missing citations you provided.
>
> -->"The empirical results could make this paper compelling, but the comparisons are rather shallow, ..."
>
> We have compared the performance of our method on CIFAR-10/100 (in Table 2) and ImageNet (in Figure 4), which are commonly used baselines in many other papers. Since RKNets are transformed from DenseNet and CliqueNet, we only compare these three models to demonstrate the effect of the transformation. To address your concern, we further added the experimental results on SVHN. Moreover, we added some related models and the experimental results with data augmentation on CIFAR. The best accuracy of DenseNet on CIFAR-100 has been also highlighted in Table 2.
>
> -->"One key reference that is missing is the careful work Chen et al https://arxiv.org/abs/1806.07366. ..."
>
> We have added the missing citation. Chen et al. (2018) utilizes a third-party package, SciPy, to provide the choices of the numerical methods and time-step sizes during both training and validation. Our work is to construct the neural network according to numerical methods directly. The benefit of our work is that the numerical methods and time-step sizes are learned and fixed after training so the computational cost in validation is little enough to offer better or similar efficiency compared to the state-of-the-art models.
>
> -->"The connections to the brain – such as ”visual cortex”, ”ventral stream”, etc – are not rigorous ..."
>
> We have removed these contents.
>
> -->"Figures 2 and 3 are the central architectural choices, but the writing around them does not clarify why ..."
>
> We have updated the paper to add more details.

---

### Official Review · AnonReviewer1 · 2018-11-09
**Extending dynamical systems view to higher order ODE solvers**

**Rating:** 4
**Confidence:** 3

**Review:**

This paper is based on an interpretation of so-called Residual Deep Networks as a discrete realization of some dynamical system. So far, work in this area used simple, Euler-forward like discretization schemes, while the paper at hand proposes to apply higher order Runge-Kutta Methods. This approach allows also to give a different interpretation of DenseNets and CliqueNets.

The general idea of this paper is interesting. Given that it is along a position paper, I think that the small set of experiments is acceptable to demonstrate the principle. What I miss however, is a clear exposition of the detailed approach. The paper lacks a lot in clarity and quality of presentation. I also think that trying to connect the presented approach to the ventral stream in the visual cortex confuses more than it helps.

---

> ### Author Response · Authors · 2018-11-26
> **Updated paper with more details**
>
> Thank you for your constructive feedback. The paper is updated with the suggested changes.
>
> -->"What I miss however, is a clear exposition of the detailed approach. The paper lacks a lot in clarity and quality of presentation. I also think that trying to connect the presented approach to the ventral stream in the visual cortex confuses more than it helps."
>
> We have updated the paper and added more details of the proposed method. Moreover, we have removed the content about ventral stream.

---

### Meta-Review · Area_Chair1 · 2018-12-17
**Nice approach, a bit more work is required**

**Confidence:** 4
**Recommendation:** Reject

**Metareview:**

The paper proposes a novel approach to neural net construction using dynamical systems approach,  such as higher order Runge-Kutta method; this approach also allows a dynamical systems interpretation of DenseNets and CliqueNets. While all reviewers agree that this is an intersting a novel approach, along the lines of recent developments in the field on dynamical systems approaches to deep nets, they also suggest to further improve the writing/clarity of the paper and also strengthen  the empirical results (currently, the method only provided advantage on CIFAR-10, while being somewhat suboptimal on other datasets, and more evidence for empirical advantages of the proposed approach would be great). Overall, this is a very interesting and promising work, and with a few more empirical demonstrations of the method's superiority as well as more polished wiriting the paper would make a nice contribution to ML community.